# Color adjustment of brand logos for dark mode display

**Byeongjin Kim**, **Giyun Lee**, **Hyeon-Jeong Suk** *

Department of Industrial Design, KAIST, Daejeon, South Korea

☉ These authors contributed equally to this work.
* color@kaist.ac.kr

## Abstract

Dark mode has become a standard feature across digital interfaces due to its visual comfort and aesthetic appeal. However, most brand logos are originally designed for light backgrounds, and when directly applied to dark backgrounds, they often suffer from color distortion, reduced visibility, and visual discomfort. These issues can negatively impact both brand identity and user experience. This study aims to propose a systematic adjustment model to optimize brand logo colors in dark mode environments. The research consisted of two experiments. In the first experiment, 31 design-major students manually adjusted 18 fictitious logos with diverse colors on a black background. The analysis revealed systematic trends in color modification, with bright colors shifting toward darker values, dark colors becoming lighter, and chroma showing an overall reduction. Additionally, red and blue hues required hue-angle corrections. Based on these findings, a convergence surface for color adjustment was constructed using Kriging interpolation, leading to the development of a predictive model applicable to new logo colors. The second experiment evaluated the model through a preference survey with 89 participants, using a set of 36 logos, including 18 fictitious logos and 18 commercial logos. Participants compared original logos with those adjusted by the proposed model. The adjusted versions were generally preferred, with the effect being particularly pronounced for logos originally featuring dark colors. The proposed model offers design principles that ensure both brand consistency and visual comfort. By integrating perceptual evidence with empirical validation, this approach provides a stable method for maintaining brand color representation in digital environments and demonstrates applicability to a wider range of graphic elements in dark mode.

## Introduction

Dark mode is an user interface (UI) mode that features white text on a dark background, which is the opposite concept of the original UI called light mode, which

**Data availability statement:** All relevant data are within the paper and Supporting information files.

**Funding:** The author(s) received no specific funding for this work.

**Competing interests:** The authors have declared that no competing interests exist.

traditionally presents black text on a white background [1]. This UI was initially introduced to reduce visual fatigue experienced by users when viewing displays, including mobile and desktop screens, in low-light environments [2]. Numerous studies have shown that the dark mode can help create a more comfortable viewing experience by reducing the contrast between the display and its surrounding environment [3–5].

Over time, dark mode has become increasingly preferred for its aesthetic appeal, particularly among younger users [6,7]. Presenting content on a dark background enhances content visibility while reducing the prominence of nonessential UI elements, creating an environment similar to a cinema screen, which enhances user immersion. As a result, major entertainment platforms such as Netflix and Spotify have adopted dark mode as a key element to appeal their brand identity [8,9]. Many companies have also integrated dark mode as a standard feature in their services [2].

Some studies have explored the impact of dark mode's background on usability. These existing studies have primarily focused on how the contrast between a dark background and white text affects reading performance and visual efficiency [10,11], aiming to determine the optimal contrast levels [3,5,12]. However, contemporary dark-mode interfaces incorporate a growing number of non-text visual elements, for which the findings of these contrast-focused studies cannot be directly applied. In particular, applying contrast optimization to brand logos often results in entirelydifferent colors being produced, which ultimately disrupts brand identity and visual recognition.

Despite this practical challenge, the appropriate perceptual adaptation of brand colors for dark mode has not been systematically investigated. Thus, a clear research gap exists regarding how brand colors should be perceptually transformed in dark environments while preserving brand identity.

### Effect of background in color perception

In general, colors are perceived differently depending on their background, meaning that even an identical color patch may appear distinct when placed against different backgrounds [13]. Since dark mode fundamentally uses a dark background, it may cause the distorted perception of colors of graphical components. Several studies have shown that changes in background color can affect perceived brightness, saturation, and hue [14,15], which highlights an important issue for interface designers to consider carefully.

To address these challenges, global corporations have introduced design guidelines for dark mode, primarily focusing on maintaining contrast between background and color elements in both light and dark modes [16–19]. These guidelines aim to ensure that the contrast between design components and a white background in light mode remains visually consistent when applied to a black background in dark mode. This approach is useful for preserving visibility and readability. However, because these existing guidelines prioritize contrast adjustments, they often result in significant color alterations. Such drastic changes can weaken or distort the brand identity conveyed through color.

Color distortions caused by background changes in dark mode can be particularly problematic in brand design. Color plays a crucial role in establishing brand identity [20,21], and logo colors are deeply tied to a company's visual identity [22–25]. Inaccurate color representation can negatively impact brand perception and recognition. However, logos are typically designed with a white background in mind [26], and applying them directly to dark mode can result in unintended color perception shifts, deviating from the designer's original intent. As UI modes change, there is an increasing need for design strategies that ensure consistent color perception across different backgrounds while preserving brand integrity.

## Considerations for logo color adaptation

The process of adjusting logo colors considered three key factors: brand constancy, visibility, and visual comfort. **Brand constancy** refers to ensuring that the adjusted color, when observed in dark mode, is perceived as the same as the original logo color typically seen in light mode. During the typical transition method to dark mode, companies have typically either retained their original logos from light mode or converted them to white for better contrast [27,28]. However, such approaches risk diminishing brand identity from a color perception standpoint. Given the symbolic role and significance of a brand's logo, any color adjustment should ensure that the originally intended color perception remains intact, preserving both brand recognition and visual consistency.

Contrast with the background significantly affects the **visibility** of UI components, including text [10,12]. For instance, placing dark-colored components on a dark mode background can result in insufficient contrast, making the components less distinguishable [11]. The same principle applies to logos. When existing logos are applied directly to dark mode without adjustment, logos with darker colors, such as those of Starbucks or Samsung, may fail to maintain sufficient visibility, potentially leading to a negative impact on brand recognition.

**Visual comfort** is strongly related to the nature of human vision, which naturally adapts to surrounding environments and background brightness [29]. For example, in a dimly lit environment, the eyes adjust to become more sensitive to light [30,31]. In this state, sudden exposure to bright light can cause discomfort [32]. The same issue can arise when directly applying logos to dark mode without modifications. For instance, logos with high brightness and saturation, such as IKEA's, may appear excessively bright against a dark mode background, causing visual discomfort and disrupting UI harmony.

To address this gap, the present study has two overarching aims that reflect both theoretical and practical perspectives. From a theoretical standpoint, we seek to understand how brand logo colors are perceptually altered in dark-mode environments and to formalize these perceptual tendencies into a quantitative model. From an applied perspective, we evaluate whether the proposed model can be effectively used in universal context by testing the user preference. Based on these findings, we further provide practical guidelines that designers and companies can adopt when implementing brand colors in dark-mode interfaces.

## Materials and methods

This study comprises two experiments. The first experiment explored how logo colors adapt when placed on a black background, simulating dark mode conditions. Based on these observations, we quantified color adjustment patterns and developed a predictive model. In the second experiment, we evaluated the model's ability to generate appropriate logo colors for dark mode automatically. A separate group of users assessed the model's performance by rating their satisfaction with the adjusted colors. The following sections outline the experimental setup and results. Fig 1 illustrates the overall workflow of this study.

### Color and design of the logo stimuli

**Color selection.** We composed a color palette to ensure a wide range of hue and tone diversity. The palette included eight chromatic colors, red, orange, yellow, green, cyan, blue, purple, and pink, along with gray. To incorporate tonal

**Fig 1. The research overview of this study.**

variations, we added darker shades of each color. In total, the palette comprised 18 colors, covering a broad color spectrum. Table 1 provides detailed information on the color palette used to generate the virtual logo stimuli.

  **Logo stimuli generated for the color adjustment experiment.** To minimize potential bias arising from participants' prior memory or familiarity with existing brands [33,34], we created fictitious brand logos specifically for this study. First, brand names were generated using ChatGPT 4.0 to avoid associations with real-world companies. Then, the visual forms of the logos were produced using Wix's logo-generation tool [35]. These initial drafts were further refined by the authors, all of whom have academic training in design. The refinement process involved typographic adjustments (e.g., kerning), geometric corrections, letterform articulation, and color assignment. Through this process, the logos were customized to serve as stylized, research-specific stimuli. Fig 2 presents the 18 logos that were used throughout the study.

**Table 1. Details of stimuli logo colors.**

| Color Name | H | S | V | L* | a* | b* | C* | h* |
|---|---|---|---|---|---|---|---|---|
| Red | 0 | 100 | 100 | 50.54 | 77.00 | 64.61 | 100.52 | 39.99 |
| Dark Red | 0 | 100 | 50 | 25.53 | 48.06 | 38.06 | 61.31 | 38.38 |
| Orange | 30 | 100 | 100 | 67.05 | 42.83 | 74.03 | 85.53 | 59.95 |
| Dark Orange | 30 | 100 | 50 | 34.51 | 23.98 | 44.82 | 50.83 | 61.85 |
| Yellow | 57 | 100 | 95 | 89.61 | −14.91 | 88.53 | 89.78 | 99.56 |
| Dark Yellow | 60 | 100 | 50 | 51.87 | −12.93 | 56.68 | 58.14 | 102.85 |
| Green | 120 | 100 | 90 | 79.64 | −79.45 | 76.69 | 110.42 | 136.01 |
| Dark Green | 120 | 100 | 50 | 46.23 | −51.70 | 49.90 | 71.85 | 136.01 |
| Cyan | 185 | 100 | 95 | 81.10 | −37.25 | −22.42 | 43.48 | 211.04 |
| Dark Cyan | 180 | 100 | 50 | 48.26 | −28.84 | −8.48 | 30.06 | 196.39 |
| Blue | 225 | 100 | 95 | 37.39 | 55.39 | −92.07 | 107.45 | 301.03 |
| Dark Blue | 225 | 100 | 50 | 18.33 | 31.74 | −55.80 | 64.20 | 299.63 |
| Purple | 270 | 100 | 95 | 38.68 | 79.94 | −89.70 | 120.15 | 311.71 |
| Dark Purple | 270 | 100 | 50 | 18.58 | 50.16 | −55.23 | 74.61 | 312.25 |
| Pink | 310 | 100 | 95 | 55.22 | 89.19 | −39.20 | 97.42 | 336.27 |
| Dark Pink | 310 | 100 | 50 | 28.46 | 55.69 | −24.57 | 60.87 | 336.19 |
| Gray | 0 | 0 | 70 | 72.94 | 0.00 | −0.01 | 0.01 | – |
| Dark Gray | 0 | 0 | 40 | 43.19 | 0.00 | −0.01 | 0.01 | – |

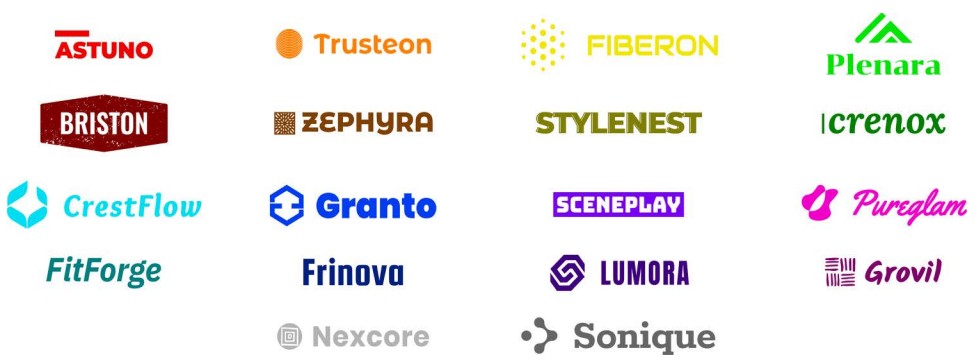

**Fig 2. 18 Fictitious logos with different colors used for color adjustment experiment.**

**Commercial logo stimuli for preference survey.** The color adjustment experiment led to the development of a color adjustment model for dark mode. To assess its applicability in design practice, we conducted an online survey featuring both the 18 fictitious logos used in color adjustment experiment and 18 additional commercial brand logos. These additional logos were selected to roughly correspond to the 18 colors in the palette, with brands such as Samsung, Starbucks, and IKEA representing dark blue, dark green, and yellow, respectively. The selected colors are not identical to those used in the previous experiment presented in Table 2, but were chosen based on similar hues to ensure the extensibility of this study.

**Table 2. Details of adjusted colors. Mean and standard deviation (SD) of the adjusted color value by the participant is provided.**

| Color Name | L* | | a* | | b* | | C* | | h* | |
|---|---|---|---|---|---|---|---|---|---|---|
| | Mean | SD | Mean | SD | Mean | SD | Mean | SD | Mean | SD |
| Red | 42.98 | 7.62 | 60.77 | 7.50 | 43.48 | 9.91 | 74.72 | 8.39 | 35.58 | 7.03 |
| Dark Red | 30.51 | 10.03 | 37.17 | 12.42 | 26.35 | 11.27 | 45.56 | 12.05 | 35.33 | 14.67 |
| Orange | 58.39 | 8.10 | 32.16 | 7.59 | 59.46 | 8.85 | 67.60 | 8.58 | 61.59 | 6.69 |
| Dark Orange | 43.84 | 12.51 | 19.37 | 8.07 | 41.18 | 11.96 | 45.51 | 11.35 | 64.81 | 11.21 |
| Yellow | 76.86 | 10.96 | −12.07 | 1.78 | 64.18 | 12.00 | 65.31 | 11.80 | 100.65 | 2.48 |
| Dark Yellow | 60.81 | 9.85 | −13.96 | 2.54 | 57.03 | 11.13 | 58.71 | 10.83 | 103.75 | 3.53 |
| Green | 68.73 | 11.29 | −61.54 | 9.43 | 55.75 | 9.65 | 83.04 | 9.53 | 137.83 | 6.59 |
| Dark Green | 52.94 | 11.56 | −49.63 | 10.92 | 45.46 | 12.43 | 67.30 | 11.63 | 137.51 | 10.02 |
| Cyan | 69.75 | 8.65 | −30.12 | 4.55 | −16.98 | 3.80 | 34.58 | 4.38 | 209.41 | 6.62 |
| Dark Cyan | 61.48 | 13.75 | −31.32 | 6.62 | −8.53 | 3.83 | 32.46 | 6.47 | 195.23 | 7.21 |
| Blue | 38.01 | 10.96 | 36.29 | 11.49 | −70.56 | 11.56 | 79.35 | 11.55 | 297.22 | 8.31 |
| Dark Blue | 32.94 | 18.85 | 27.53 | 11.76 | −56.66 | 20.07 | 62.99 | 18.77 | 295.91 | 12.50 |
| Purple | 37.53 | 8.09 | 59.19 | 11.74 | −66.86 | 13.15 | 89.30 | 12.55 | 311.52 | 7.94 |
| Dark Purple | 32.80 | 16.16 | 49.41 | 14.37 | −56.64 | 16.65 | 75.16 | 15.71 | 311.10 | 11.74 |
| Pink | 46.24 | 9.00 | 65.69 | 12.59 | −31.30 | 5.48 | 72.77 | 11.61 | 334.52 | 5.78 |
| Dark Pink | 35.68 | 13.60 | 46.51 | 12.60 | −22.76 | 7.80 | 51.78 | 11.83 | 333.92 | 9.88 |
| Gray | 64.03 | 12.22 | 0.35 | 1.60 | 0.31 | 0.92 | 0.47 | 1.34 | 41.53 | 55.00 |
| Dark Gray | 62.41 | 14.98 | 0.37 | 1.94 | 0.53 | 1.37 | 0.65 | 1.58 | 55.08 | 57.21 |

## Participants

**Participants of color adjustment experiment.** A total of 31 Korean design-major students familiar with digital design tools including Figma and Adobe Illustrator, including 19 females and 12 males, participated in the color adjustment experiment. Their mean age was 24.32 years with standard deviation of 2.62. All participants had normal color vision confirmed using the Ishihara 38-plate test prior to the experiment and received $20 USD for their voluntary participation. Since the experimental materials were prepared on the Figma platform, proficiency in Figma was required. Participants for this study were recruited through university mailing lists and departmental bulletin boards between July 28 and July 30, 2025. No participants were excluded or withdrew from the study.

Participants were provided with a consent form that included detailed information about the study purpose, procedures, potential risks and benefits, voluntary participation, compensation, and data privacy measures. They were given sufficient time to read and understand the document before signing, and the original written consent forms were securely stored in accordance with Institutional Review Board (IRB) regulations (KAIST IRB; approval No. KAISTIRB-2025–183). The consent process wasconducted individually prior to the start of the study, and participation was permitted only after the researcher confirmed that each participant had fully understood the information.

**Participants of preference survey.** In the preference survey, we recruited a separate group of college students. A total of 89 Korean participants took part in the online test, including 36 females and 53 males. Their mean age was 23.93 years with standard deviation of 3.15. Unlike the color adjustment experiment, participation was not restricted to individuals with a design-related background. All participants had normal color vision and received $10 USD for their voluntary participation. The experimental sessions were subsequently conducted from August 13 to August 15, 2025 through university online community.

For the online survey, electronic informed consent approved by an identical IRB protocol was obtained on the opening screen, and participants were required to agree before proceeding. This consent was electronically documented through the survey system's time-stamped logs. No direct personal identifiers were collected. Survey records and laboratory data were stored only accessible to the research team, and the analytic dataset used for this manuscript consisted of anonymized records. To verify low-light conditions for dark-mode usage, participants were asked to upload a photo of their ambient environment. The images included in the supplementary materials did not contain identifiable individuals, and participants were instructed to ensure that no elements revealing personal identity or household location were visible.

## Procedures

All participants were adults, and no minors were enrolled. Written informed consent was obtained from participants in the Color Adjustment Experiment, and electronic informed consent was documented via the survey platform in the Preference Survey. No parental or guardian consent was applicable, and no consent waiver was requested or granted.

**Procedure of color adjustment experiment.** The experiment was conducted in a low-light environment with an illuminance below 1 lx to simulate typical dark mode usage in dim settings while minimizing the influence of ambient lighting and reflections. Before the experiment began, participants were given approximately five minutes for light adaptation.

The experiment was conducted on a Samsung Galaxy Book 3 with 1920×1080 resolution. The experimental monitor's automatic brightness adjustment was activated, which maintained display brightness at 40% of the maximum (100%) throughout the experiment. Using a Spectroradiometer CM2000, we confirmed that the average luminance of a full-screen white background was 49.26 cd/m2, validated as an appropriate brightness level [32,36]. The monitor's white balance was set to D65, the default display configuration, and remained unchanged.

For the color adjustment task, Figma was used in dark mode, providing an interface optimized for color modification. As shown in Fig 3, participants adjusted colors using Figma's HSB (Hue, Saturation, Brightness) interface. During

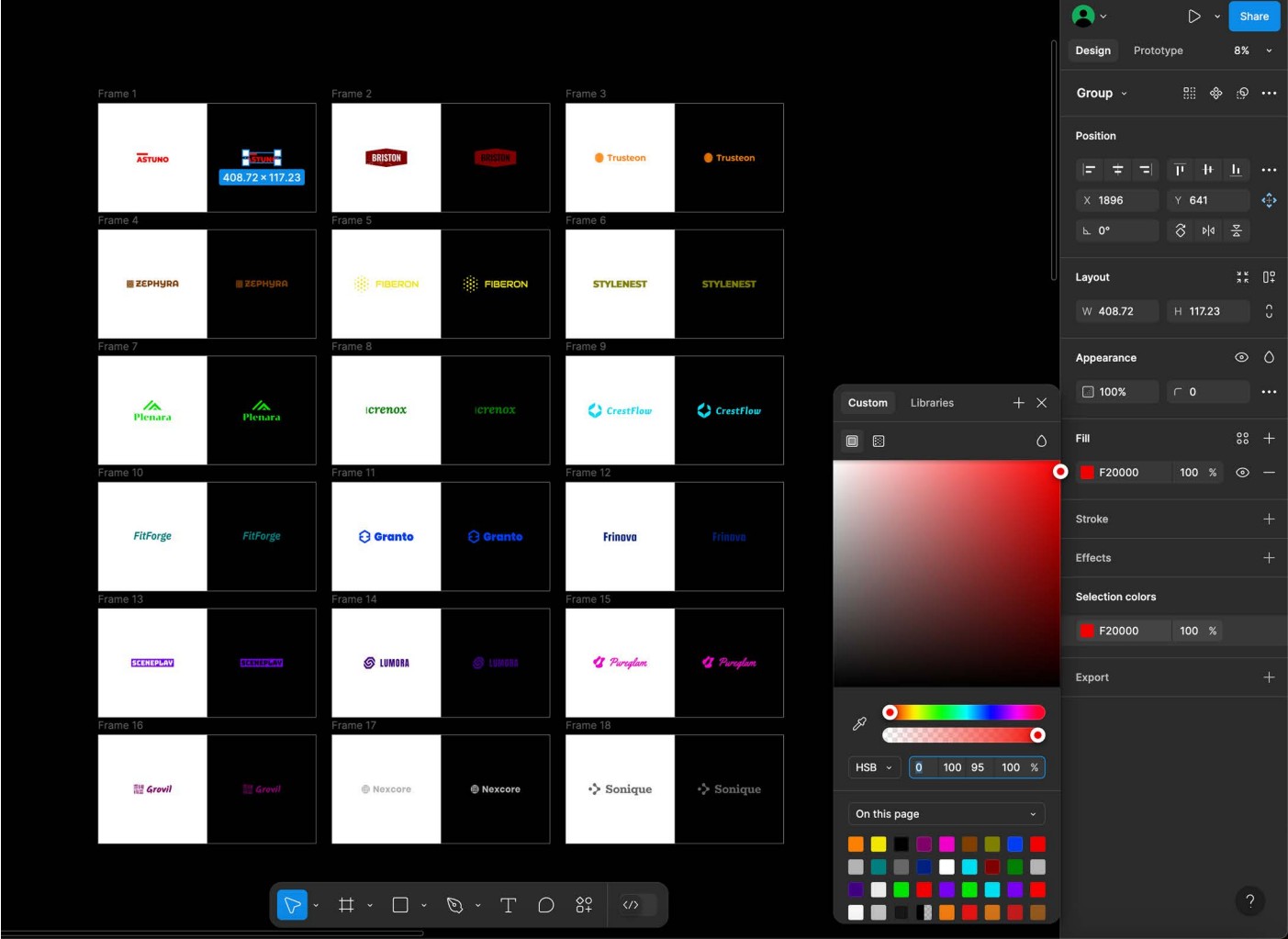

**Fig 3. The initial Figma workplace provided to participants for the color adjustment experiment.** When a participant began the experiment, the 18 logo pairs were presented in white (left) and black (right) background. While minimizing the loss of the color characteristics of the brand logo, participants adjusted color of the logos in black background aiming at visual comfort, visibility, and aesthetic appeal. Most of the participants zoomed the screen during the color adjustment.

the experiment, a color space that enabled intuitive color adjustments was preferred to enhance user interaction. HSB color model is functionally identical with HSV (Hue, Saturation, Value) model, and they are often used interchangeably in graphic design tools, including Figma [37]. In HSV, H (Hue) represents color and is expressed as an angular rotation along the central axis. S (Saturation) represents color vividness and is defined as the distance from the neutral axis. V (Value) denotes brightness and is defined along the central axis.

At the start of the experiment, each of the 18 fictitious logos was presented as a pair, with one displayed on a white background and the other on a black background within the Figma workspace. The logo on the white background represented the original design in light mode and served as a reference for participants when adjusting the logo color for dark mode. While some degree of color adjustment was expected, preserving the brand's color identity remained a key requirement. Participants were instructed to adjust the logo's color on the black background while maintaining its perceived color identity from the white background while enhancing visual comfort and visibility.

**Procedure of the preference survey.** The preference survey was conducted online, and participants were instructed to complete it in a dark environment to align with typical dark mode usage. Before starting, they were asked to dim their surroundings and activate the automatic brightness adjustment and white balance settings on personal mobile devices to ensure appropriate display conditions. Additionally, they were required to submit a photograph of their surroundings to assess the practicality of the adjusted logo colors under real-world dark mode conditions. Some example photographs are presented in Fig 4.

During the survey, participants were first presented with the original logo on a white background, serving as a reference for identifying the brand's aesthetic characteristics. Below this, two variations of the logo on a black background were displayed. As shown in Fig 5, the left-side logo retained its original color, whereas the right-side logo featured the adjusted color recommended by the color adjustment model. Participants compared the two variations while evaluating whether the adjusted logo maintained its color identity relative to that perceived on the white background, and instructed to select the preferred version.

A total of 36 logos, including 18 fictitious logos and 18 commercial brand logos, were presented in randomized order. Additionally, participants completed a familiarity check, indicating whether they recognized each logo. This familiarity check forecasted color memory influenced participants' tolerance for color changes [25].

## Results

Based on the observed color adjustments, we developed a model that describes how logo colors adapt and converge in dark mode. The following sections present the findings, model construction, and validation results.

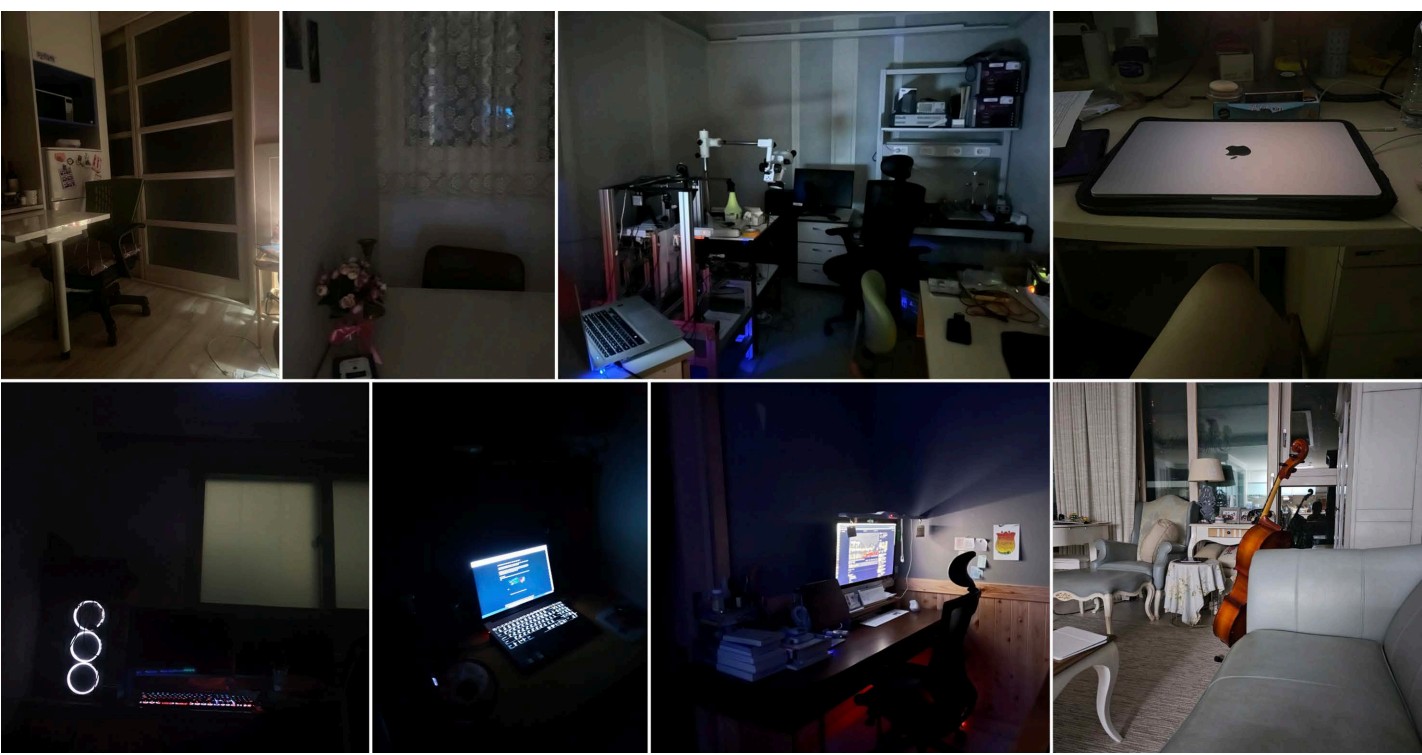

**Fig 4. The examples of participants' surroundings in preference survey.** We asked participants to join the survey in a dark setting. Since the test was carried out remotely, we requested to upload their surrounding images before answering the survey questionnaires.

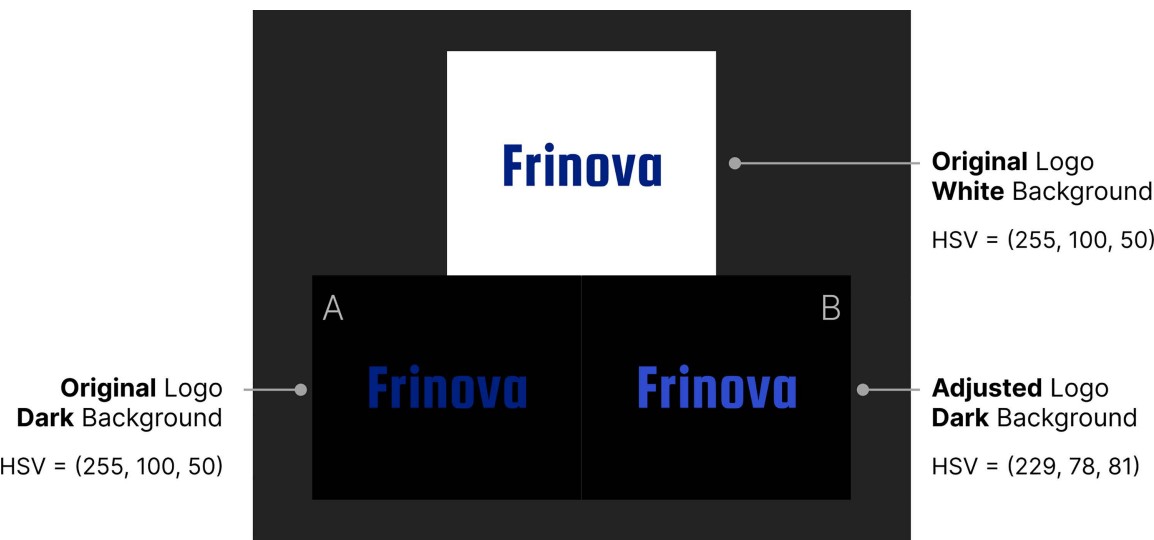

**Fig 5. Example of a comparative evaluation.** The top image shows the brand's original logo on a white background. The bottom-left image displays the original logo on a black background, while the bottom-right image presents the color-adjusted logo on a black background.

For color representation, as previously mentioned, color selection and adjustments were performed using the HSV color system, as it aligns with the intuitive workflow of design professionals. However, HSV has perceptual limitations, as Euclidean distances in this space do not necessarily correspond to actual perceptual differences [38]. Additionally, Saturation (S) and Value (V) are not fully independent attributes, as variations in one can affect the perception of the other. For example, when $V = 0$, the color appears black, regardless of changes in S. Despite these drawbacks, HSV remain widely used in graphic design tools, making it a practical choice for presenting experimental stimuli in the color adjustment task.

For quantitative color analysis, we converted HSV color inputs to the CIE1976 L*a*b* (CIELAB) color space to better reflect human color perception [38]. CIELAB provides a perceptually uniform color space, enabling a more meaningful interpretation of color differences. CIELAB is a perceptually uniform color space, meaning that equal distances between colors in its L*a*b* dimensions correspond to approximately equal perceived color differences (ΔE*) [38]. This makes it well-suited for modeling perceptual color shifts. Notably, as color adjustments in dark mode primarily affected Chroma and Lightness, we further transformed CIELAB values into CIE1976 L*C*h (Lightness, Chroma, Hue, CIELCh). These properties allowed us to systematically analyze color adjustments and derive principles for perceptually meaningful color transformations in dark mode.

### Trend of color adjustments for dark mode display

Based on 31 design-major students' color adjustments, we estimated the average values color attributes that are comparatively presented as shown in Fig 5. Overall, the adjusted colors for the dark mode became more grayish and less saturated. Bright colors became darker, whereas dark colors became brighter. It implies that bright and highly vivid logo colors shall appear less bright and less vivid in black background. On contrast, dark logo colors need to be brightened to be more visible in black background. Through observations, we tried to build a model by incorporating the color attributes. Table 2 and Fig 6 shows the result of color adjustment experiment, while detailed raw data are provided in S1 Table.

- **L\* (Lightness)**: In the color adjustment experiment, the palette was designed with two brightness levels for each of the eight hue categories, forming color pairs. The results showed that lighter colors within a pair tended to decrease in

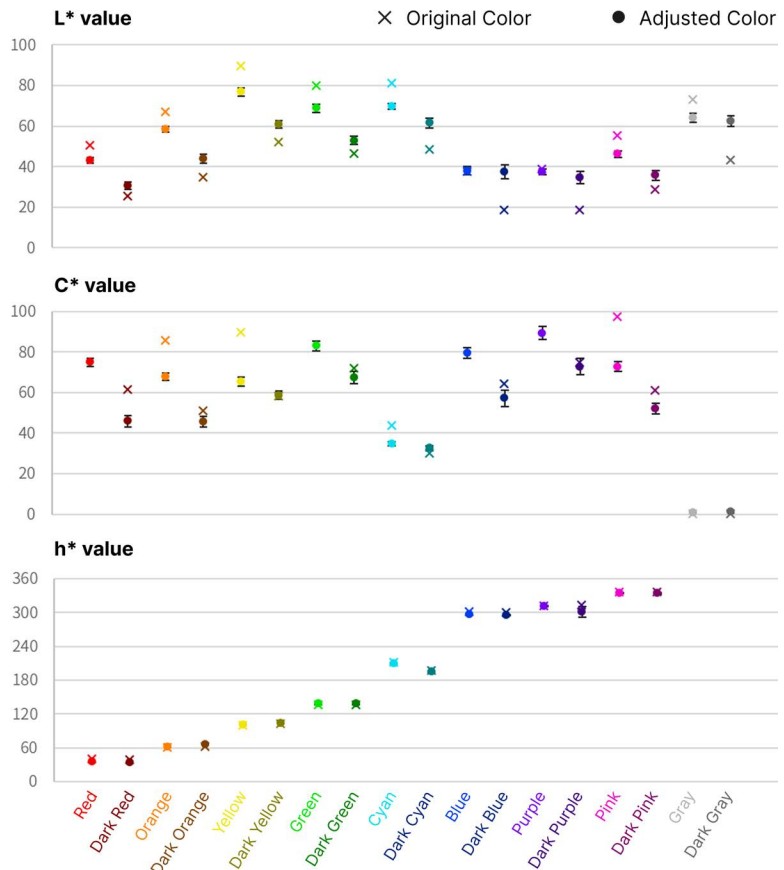

**Fig 6. The comparison plot between adjusted logo color result with mean and standard error and original logo color.**

L* by average of 7.74 (SD = 7.27), while darker colors increased in L* by average of 11.65 (SD = 10.58). The difference between the two groups was statistically significant (t(30) = −16.98, p < .01). The greater the difference between the original and the optimal L* value, the larger the observed brightness adjustment. For instance, yellow (L* = 80.0) exhibited an average reduction of approximately 20.0, whereas purple (L* = 40.0) showed only a minor adjustment less than 2.0. This indicates that hue should be considered as a variable when modeling optimal logo colors for dark mode.

- $\sqrt{a^{*2}+b^{*2}}$ **(Chroma, C*):** Colors with higher chroma values exhibited a greater reduction in C*, whereas colors with lower chroma values underwent minimal adjustment, supported by a Spearman rank correlation analysis which revealed a very strong negative association between chroma changes in bright and dark color groups (ρ = −0.88, p < .01, N = 18).

- $tan^{-1}\left(\frac{b^{*}}{a^{*}}\right)$ **(Hue, h*):** With the exception of red and blue, little to no variation remained below 4° in hue was observed. The process of adjusting logo colors on a black background resulted in bright colors becoming darker and dark colors becoming brighter, while the hue angle remained relatively stable. However, the hue angles of red and blue exhibited slight reductions, suggesting that human perception of these hues interacts with their chroma characteristics. This tendency was previously reported by Hung and Berns [39], who claimed that hue angle adjustments to maintain perceptual color consistency as chroma changes. Their dataset provided coordinate mappings of red and blue along chroma variations. The empirical findings from our color adjustment experiment replicated this trend, and therefore, we incorporated this hue rotation into the development of the color adjustment model for dark mode.

These observations provide insights into how the convergence points of different colors, as well as their positions in color space, can be incorporated into the development of a color adjustment model.

## Building a color adjustment model for dark mode

The color adjustment model aimed to define the convergence area where dark mode-adjusted colors tend to settle. We deployed 36 colors in the CIELAB system, including the initial 18 colors as well as the corresponding 18 adjusted colors. The coordinates for the adjusted colors were derived by averaging the values obtained from 31 participants.

Specifically, we sought to identify an irregularly curved surface and examine the relationship between the initial color, the adjusted color, and the projection of their connection on the surface. The detailed modeling process is described in the following step-by-step procedure.

1. **Defining and Smoothing the Convergence Surface for Color Adjustment**

The results of the color adjustment experiment showed that bright colors tended to be adjusted toward a darker direction, while dark colors were adjusted toward a brighter direction. During this process, we observed that the adjusted colors converged toward a specific hypothetical layer. Based on this observation, we interpolated this layer to establish a reference for color adjustment convergence. To achieve this, adjusted color points with the same hue category were paired, resulting in a total of nine adjusted color pairs. The midpoint of each pair was then calculated, and this midpoint was assumed to lie on the color adjustment convergence surface. The calculations were conducted in CIELAB space.

To generate the convergence surface, we applied the Gaussian variogram model and Ordinary Kriging interpolation [40,41]. This interpolation method models the spatial correlation between data points using a variogram and estimates values at unobserved locations based on the weighted average of neighboring data points. Among various interpolation methods, Kriging was selected because it provides statistically optimal predictions under second-order stationarity, yields a smooth and continuous surface without overfitting local fluctuations, and allows variance estimation for each prediction, which is essential for assessing model reliability [40]. The Gaussian variogram provides a continuous and differentiable spatial process, reflecting the gradual and perceptually smooth transitions of color shifts in the CIELAB space [42]. These enabled the generation of a convergence surface that avoids abrupt changes or unrealistic oscillations even in sparsely sampled regions. The smooth convergence surface is shown in Fig 7 provides an alternative perspective of the surface.

2. **Computing Relative Position to the Convergence Curve**

The adjustment vector represents the change in color from its original state, as observed in the previous color adjustment experiment, and is expressed as a vector in the CIELAB color space. This vector can be relatively computed with respect to the derived convergence surface. To achieve this, the adjustment vector is expressed in relation to the convergence surface and decomposed into its respective components.

To analyze this relationship, the plane that includes both the original color and the $L^*$ axis is identified, and its intersection with the convergence surface is determined. This intersection forms a curve in the $L^*$-$C^*$ plane. Since $C^*$ represents chroma and is defined as the Euclidean distance from the origin in the ab plane, this curve is positioned within the $L^*$-$C^*$ plane.

In the $L^*$-$C^*$ plane, the relative displacement of the original color with respect to the curve is quantified by two key distance measures. A perpendicular foot is drawn from the original color to the curve, defining two displacement components:

- $d_{norm}$: The perpendicular distance from the original color to its foot on the curve

- $d_{curve}$: The tangential distance along the curve from its intersection with the L axis to the perpendicular foot

Thus, $d_{norm}$ is assigned a negative sign when the original color is positioned below the curve and a positive sign when it is above the curve. These two displacement values represent the relative position of the original color with respect to the curve, providing a quantifiable measure of its deviation in both the perpendicular and tangential directions.

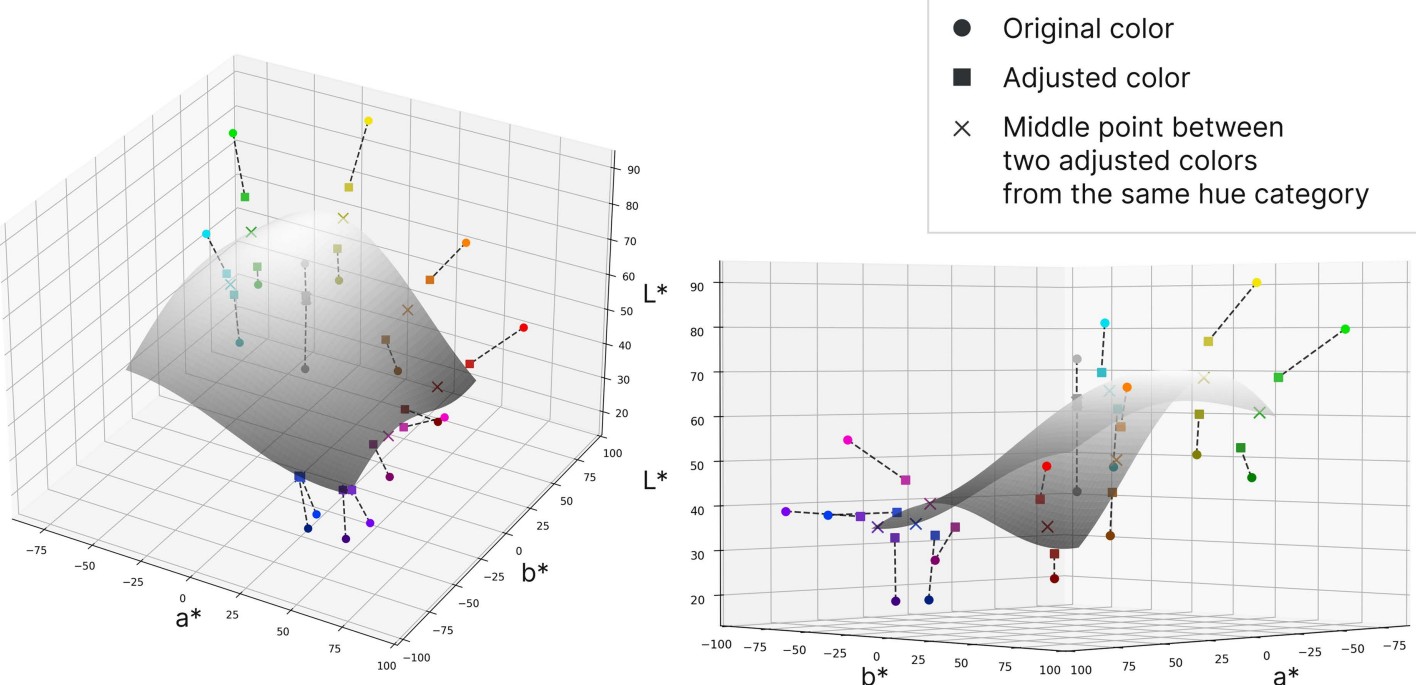

**Fig 7. Convergence surface for color adjustment.** Circles represent the 18 initial colors, while squares represent the adjusted colors. Within the same hue category, brighter colors became darker, and darker colors became brighter, while remaining within the intermediate layer area. The curved surface passes through this area.

The adjustment vector consists of components expressed in CIELAB, which can be converted into CIELCh. While it inherently contains directional components in $L^*$, $C^*$, and $h^*$, the $h^*$ component requires a separate adjustment process. Therefore, the analysis first focuses on changes within the $L^*$-$C^*$ plane. The relationship between the adjustment vector and the derived curve is examined by establishing two new orthogonal axes: one in the perpendicular direction and the other in the tangential direction.

- $d^*_\perp$: The perpendicular displacement component of the adjustment vector
- $d^*_\parallel$: The tangential displacement component of the adjustment vector

Thus, $d^*_\perp$ is assigned a positive sign when moving toward the convergence curve and a negative sign when moving away from it. Similarly, $d^*_\parallel$ is assigned a negative sign when directed toward the $L^*$ axis and a positive sign when moving away from it. The adjustment vector has components along these two axes, and by decomposing these components, the relative displacement with respect to the curve can be defined. Fig 8 illustrates the procedure.

### 3. Deriving Changes in Chroma and Lightness

Using the variables derived in the previous steps, it is possible to calculate the adjusted color for a new arbitrary logo color to be applied in dark mode. To achieve this, a quadratic regression analysis was conducted to develop a model that predicts the intensity and direction of color adjustment.

First, the regression model for predicting $d^*_\perp$ was constructed using dnorm, while the model for predicting $d^*_\parallel$ was derived based on $d_{curve}$. The corresponding regression equations are as follows:

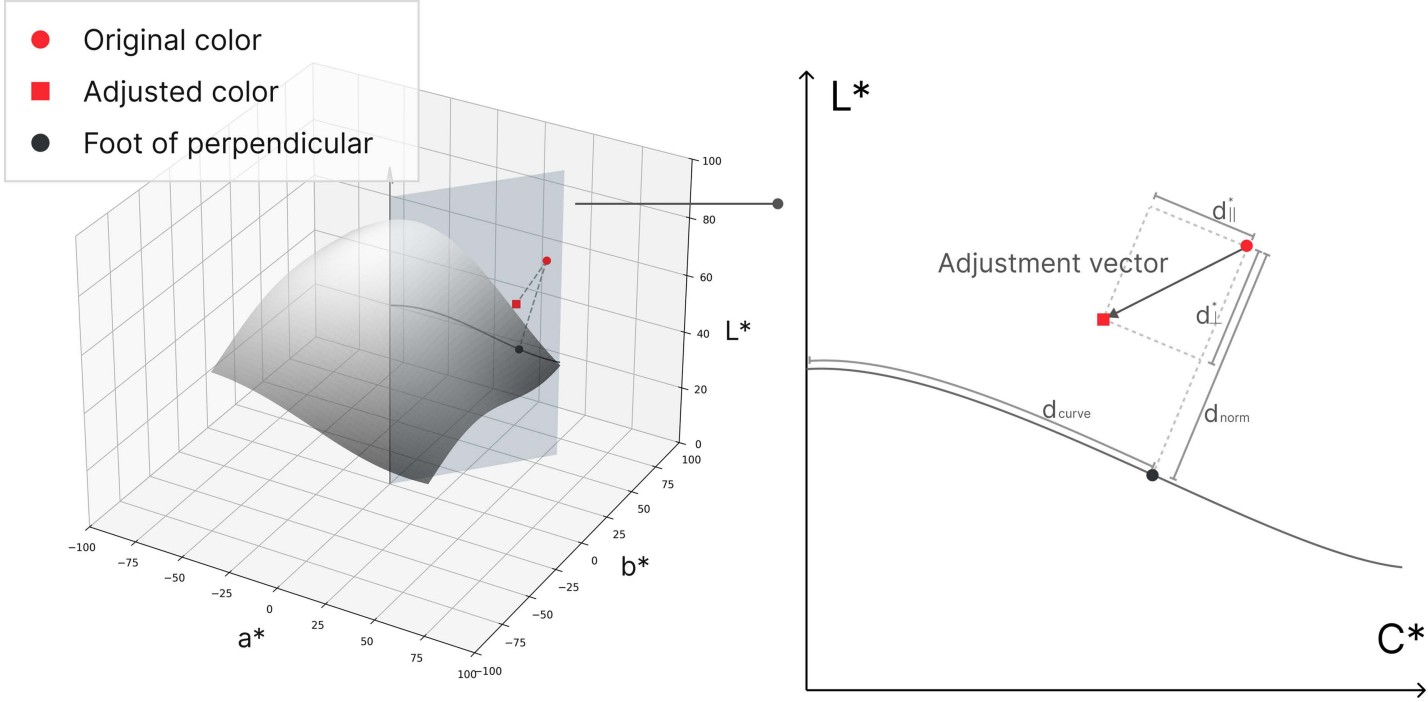

**Fig 8. Computing relative position of adjustment vector and original color.** Through this process, the variable $d_{norm}$, $d_{curve}$, $d^*_{\perp}$, $d^*_{\parallel}$, which represents the relative position with respect to the curve, was defined.

$$d^*_{\perp} = 9.47 \times 10^{-4} \times d^2_{norm} - 5.92 \times 10^{-1} \times d_{norm} \ \left(R^2 = .90\right) \tag{1}$$

$$d^*_{\parallel} = -2.60 \times 10^{-3} \times d^2_{curve} - 5.56 \times 10^{-2} \times d_{curve} \ \left(R^2 = .91\right) \tag{2}$$

The derived regression equation 1 indicates that as the original color deviates further from the convergence surface, the magnitude of the perpendicular displacement increases. This implies that a greater adjustment is required toward the convergence surface. Similarly, regression equation 2 suggests that the farther a color is from the $L^*$ axis, the greater the adjustment required in the direction of the $L^*$ axis. In other words, colors with higher chroma tend to experience a more significant reduction in chroma during the adjustment process.

When a new original logo color is given to the model, we can calculate the values of $d_{norm}$ and $d_{curve}$ for the color point $O$ using the previously described process. Based on these values, the regression models derived earlier can be applied to determine the adjusted displacement values $d^*_{\parallel}$ and $d^*_{\parallel}$ required for the logo color $O$ to be adapted for dark mode. Finally, these adjusted displacement values allow us to compute the $L^*$ and $C^*$ values for the adjusted color $O'$.

### 4. Correcting the Hue Angle of Adjusted Color $O'$

During the logo color adjustment process, participants consistently exhibited a tendency to reduce chroma. However, previous studies on color perception have shown that for red and blue, maintaining perceptual color consistency requires simultaneous adjustments to the hue angle when modifying chroma [39,43]. When reducing the chroma of red, the hue should shift slightly toward purple, while for blue, it should shift toward green. This adjustment ensures that the perceived color identity remains consistent with the original high-chroma reference red and blue. A similar tendency was observed

in the result of the color adjustment experiment. Moreover, [39] offers a table of references that describe how hue angle changes along the Chroma varies. Accordingly, we consult hue angle adjustments based on the Look-Up Table (LUT) provided by Hung and Berns [39], which quantifies these perceptual shifts. The Table 3 shows the LUT utilized for the hue adjustment.

To be specific, hue angle correction was performed by referencing the two closest reference colors to the target color O and calculating the relative angular distance between them. The detailed methodology is described as follows:

**(a) Reference Color Selection**

Rather than covering the entire color space, the LUT consists of a set of representative sample points. Specifically, the table defines reference chroma levels at 1/4, 2/4, 3/4 and 4/4 (Ref) of maximum Chroma, structuring them into a segmented interpolation model. Given a target color *O*, the two closest reference colors are selected based on their hue proximity within this segmented space. To systematically adjust hue, the original color *O* and the adjusted color with undefined hue *O′* are mapped onto the a*-b* plane. Circles are drawn with radii corresponding to their distances from the origin.

**Table 3. Look-up table used for hue angle Correction [39].**

| Chroma | L | C | h | | L | C | h |
|---|---|---|---|---|---|---|---|
| Red | | | | Red-yellow | | | |
| ¼ | 62.42 | 27.44 | 29.71 | | 82.58 | 14.32 | 65.91 |
| 2/4 | 62.42 | 52.90 | 36.25 | | 82.58 | 33.89 | 70.89 |
| ¾ | 62.42 | 77.82 | 37.60 | | 82.58 | 57.26 | 74.15 |
| Ref | 62.42 | 109.81 | 45.31 | | 82.58 | 89.46 | 75.27 |
| Yellow | | | | Yellow-green | | | |
| ¼ | 100.16 | 13.42 | 102.78 | | 97.92 | 16.26 | 132.96 |
| 2/4 | 100.16 | 34.93 | 102.52 | | 97.92 | 40.57 | 125.37 |
| ¾ | 100.16 | 60.81 | 102.69 | | 97.92 | 68.16 | 121.95 |
| Ref | 100.16 | 98.60 | 99.81 | | 97.92 | 104.72 | 117.98 |
| Green | | | | Green-cyan | | | |
| ¼ | 90.62 | 23.74 | 151.28 | | 93.78 | 12.11 | 187.30 |
| 2/4 | 90.62 | 55.14 | 143.07 | | 93.78 | 29.96 | 180.42 |
| ¾ | 90.62 | 89.52 | 140.37 | | 93.78 | 48.39 | 179.00 |
| Ref | 90.62 | 128.49 | 139.73 | | 93.78 | 68.68 | 176.34 |
| Cyan | | | | Cyan-blue | | | |
| ¼ | 88.10 | 11.69 | 233.00 | | 73.06 | 15.47 | 263.03 |
| 2/4 | 88.10 | 23.93 | 217.63 | | 73.06 | 26.86 | 249.62 |
| ¾ | 88.10 | 37.25 | 212.93 | | 73.06 | 38.88 | 245.64 |
| Ref | 88.10 | 52.21 | 207.55 | | 73.06 | 51.08 | 249.38 |
| Blue | | | | Blue-magenta | | | |
| ¼ | 39.19 | 25.46 | 279.18 | | 55.38 | 29.84 | 316.43 |
| 2/4 | 39.19 | 47.92 | 281.59 | | 55.38 | 56.00 | 317.98 |
| ¾ | 39.19 | 74.20 | 292.51 | | 55.38 | 81.48 | 317.25 |
| Ref | 39.19 | 122.79 | 303.41 | | 55.38 | 110.64 | 317.38 |
| Magenta | | | | Magenta-red | | | |
| ¼ | 67.15 | 29.91 | 326.75 | | 66.16 | 27.00 | 343.68 |
| 2/4 | 67.15 | 54.22 | 329.97 | | 66.16 | 49.40 | 343.68 |
| ¾ | 67.15 | 78.94 | 330.87 | | 66.16 | 68.78 | 346.72 |
| Ref | 67.15 | 106.00 | 328.50 | | 66.16 | 86.49 | 349.12 |

**(b) Computing Reference Color Hue Angle Correction**

The reference color segments from the LUT intersect the circles at four points, two intersection points for *O* before adjustment, and two intersection points for *O′* after chroma and lightness adjustment. The relative positions of *O* and *O′* between these reference colors define their hue shift behavior. To compute the hue correction, the hue shift values for the two reference colors are obtained. These values represent the expected hue adjustments when transitioning from the original reference hues to the adjusted state.

**(c) Applying Inverse Distance Weighted (IDW) Interpolation**

Using the initial hue difference between *O* and its reference colors, we apply the Inverse Distance Weighted (IDW) interpolation method. The computed hue shift is applied to *O′* to determine the final adjusted hue angle.

This ensures that colors retain perceptual consistency in dark mode while minimizing hue distortion. Fig 9 shows the illustration of the hue angle correction process. To enhance transparency and reproducibility, the Python implementation of the color adjustment model, along with supplementary CSV files and a Jupyter notebook demonstrating the simulation workflow, is publicly available in S1 Code and Zenodo: https://doi.org/10.5281/zenodo.16939055

**Performance test from preference survey**

A total of 86 preference responses were collected through an online survey. The results showed that, except for the Starbucks logo, more than 50% of participants preferred the adjusted colors. This finding indicates that the proposed color adjustment model functions effectively for logo color adaptation in dark mode, demonstrating its practical applicability. Fig 10 presents the proportion of users who preferred the adjusted logo, while the raw preference data used for the analysis are provided in S2 Table.

The results of this experiment allow for further analysis of the factors influencing user preferences for adjusted logo colors. User memory colors for logos may influence their preferences for adjusted color versions. To examine this relationship, a chi-square test of independence was conducted to determine whether logo familiarity was associated with preference for the adjusted version. The association was statistically significant ($\chi^2(1) = 10.11$, $p < .01$). However, the effect size was small (Cramer's $V = .08$), indicating that although familiarity and preference are related, the strength of this association is weak. Thus, participants who reported recognizing a logo were slightly more likely to prefer its original color rather than the adjusted version, suggesting that pre-existing memory color acts as a weak but measurable anchor in their judgments.

Additionally, due to the characteristics of the proposed model and its color adjustment process, the adjustment magnitude is greater when the initial color of a logo is farther from the convergence surface, whereas colors closer to the surface tend to retain their original appearance even after adjustment. To examine the relationship between color variation and user preference, we calculated the Euclidean distance in the CIELAB space between the original and adjusted colors and compared it to the proportion of users who preferred the adjusted logo. The Spearman's correlation analysis resulted in $\rho = -.13$ ($p = .46$), indicating no significant correlation between the two variables. This suggests that a larger color adjustment does not necessarily lead to a decrease in user preference.

## Discussion

### Interpretation of findings

This study empirically analyzed how brand logo colors perceptually shift in dark mode environments and whether such changes exhibit consistent, predictable patterns. Based on these findings, we developed a color adaptation model that enables designers to apply brand colors in dark mode while preserving visual identity.

The observed color adjustment tendencies are not merely a matter of aesthetic preference. Rather, they are directly linked to how the human visual system processes color and contrast information in low-luminance environments [32].

(a) Reference Color Selection

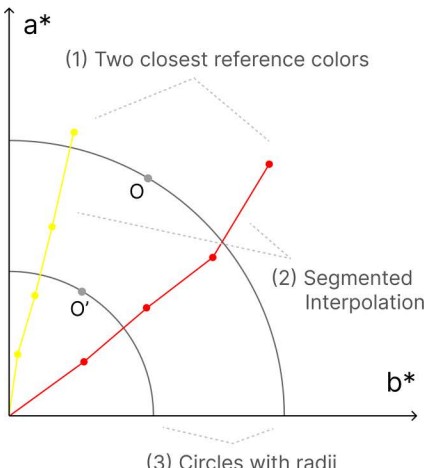

(b) Computing Reference Color
Hue Angle Correction

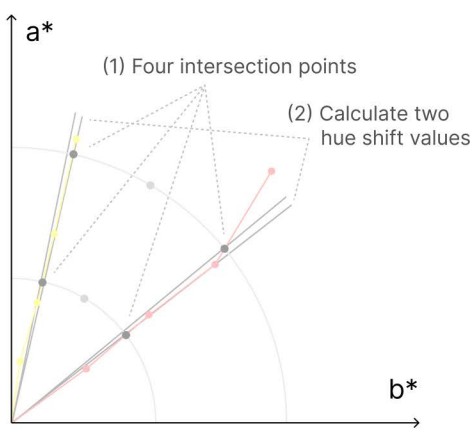

(c) Applying Inverse Distance Weighted Interpolation

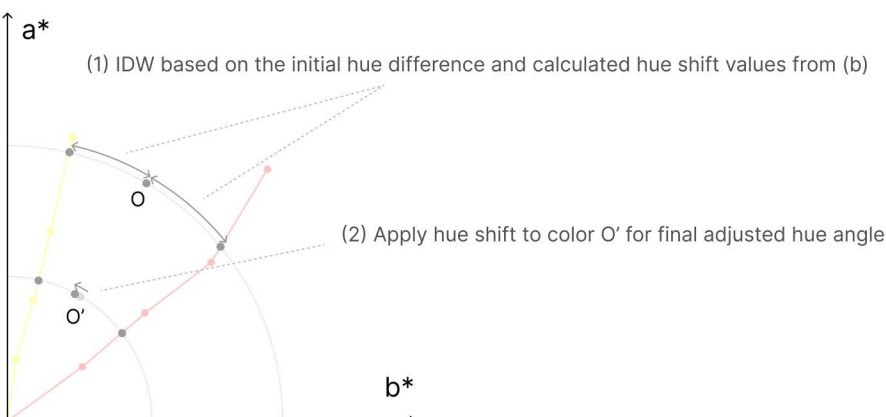

**Fig 9. Hue angle correction process for adaptation.** The figure consists of three main steps: **(a)** Reference color selection, **(b)** Computing reference color hue angle correction, and **(c)** Applying Inverse Distance Weighted interpolation.

While previous dark mode research has primarily focused on text contrast and readability [3,5], the present study demonstrates that the same perceptual mechanisms apply to brand colors and logo design. However, unlike text, logo colors must maintain brand constancy [44], thus requiring a distinct approach. This study addresses this need by proposing a perceptually grounded method for dark mode color adaptation that balances both consistency and comfort.

The characteristic tendency in dark mode, where lighter colors appear dimmer and darker colors become relatively more prominent, can be explained by Weber–Fechner law [45] and contrast normalization [46]. Rather than responding to absolute luminance, the human visual system is more sensitive to relative differences and background contrast. Excessive contrast can induce visual fatigue, prompting perceptual compensation toward more neutral contrast levels. The findings of this study provide new empirical evidence that this contrast-adaptation mechanism applies not only to textual elements but also to logos and brand color systems.

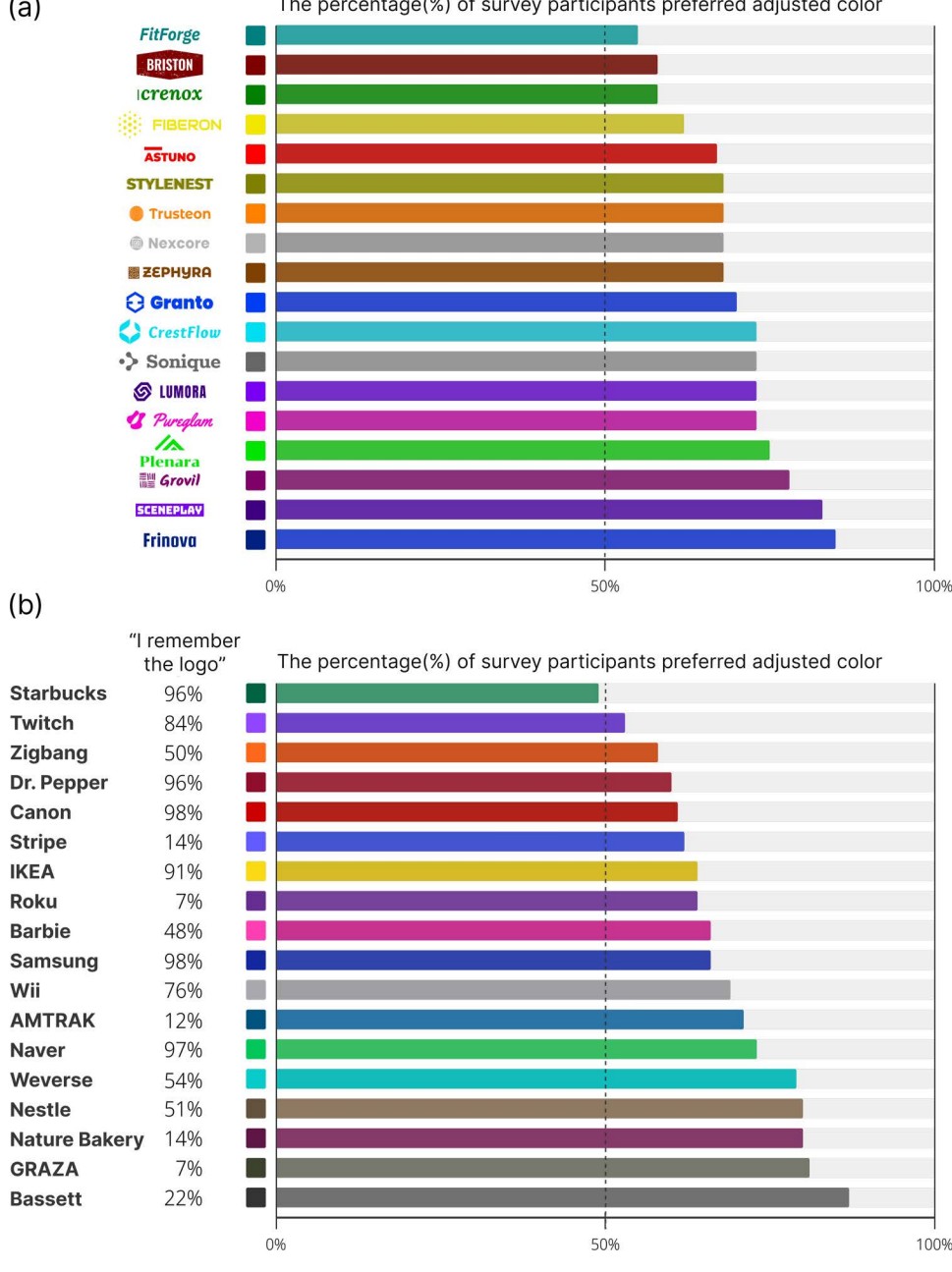

**Fig 10. Proportion of users preferred adjusted logo.** An online survey evaluated 36 logo stimuli, with **(a)** 18 fictitious logos and **(b)** 18 commercial brand logos. The majority of participants preferred the adjusted colors when logos were presented on a black background in dark mode.

The finding that hue remained largely stable across dark mode transformations aligns with color psychology theories on memory color. Prior studies have shown that even when overall color appearance changes, the preservation of hue is crucial for consistently recognizing familiar objects [47,48]. This principle is particularly relevant for brand logos, whose recognition relies heavily on stable hue memory [25]. Accordingly, the present study provides perceptual evidence that maintaining hue is a key strategy for preserving brand identity in dark mode environments [39,43].

Furthermore, the observed color shifts were not random or device-dependent RGB changes, but instead formed continuous and spatially correlated patterns in the CIELAB color space. This suggests that color adaptation in dark mode can be modeled as a transformation field within a perceptual space rather than as discrete numerical adjustments. This forms the theoretical basis for employing Kriging-based spatial interpolation to construct the convergence surface and supports the broader notion thatdigital branding colors can be represented as perceptual functions rather than as isolated RGB values.

## Comparison with previous studies

Most prior research on dark mode or negative polarity has focused primarily on text perception and reading performance [10,11]. These studies typically examined how recognition of characters changes under different combinations of background and text luminance and consistently concluded that a minimum level of luminance contrast is required to ensure legibility and visual efficiency [49].

In essence, previous work has emphasized contrast-preserving strategies, maintaining sufficient luminance contrast between foreground and background when switching the interface polarity. However, very few studies have explored how hue or saturation should be perceptually adapted in dark mode, nor have they provided quantitative models for such adjustments. This gap is even more evident in the context of brand logos, where a single color is directly tied to brand identity. Despite the importance of color constancy, research has rarely addressed how brand colors should be preserved or transformed in dark environments, which highlights the novelty of the present study.

Some researchers have also shown that ambient illuminance and display type significantly influence text visibility and visual fatigue [50]. These findings suggest that the effects of dark mode are not determined by a single factor but emerge from the interaction of user characteristics, task type, ambient lighting, display luminance, and contrast polarity [51]. In the present study, however, the display device and ambient lighting conditions were tightly controlled to isolate the perceptual mechanism of color adaptation. This decision aligns with the methodological characteristics of this research but also indicates a potential direction for future work, where different devices, display technologies, and lighting environments should be systematically compared to further validate the proposed model.

## Practical design implications

From an industry perspective, brands typically adopt two main strategies when applying their logos in dark mode environments. The first is a contrast-preserving strategy, exemplified by brands such as Netflix [8]. In this approach, the original logo colors are retained, while the background is constrained to ensure sufficient luminance contrast. This method effectively maintains brand recognition, but the logo can only be used against a limited range of backgrounds, which restricts its visual adaptability across different contexts. In contrast, platforms like Spotify employ a monochrome strategy, using a white logo in dark mode [9]. This approach harmonizes with the overall dark UI atmosphere and preserves emotional tone, but it sacrifices color-based brand communication, resulting in a loss of chromatic identity.

Thus, the color adjustment model proposed in this study provides a systematic and reproducible framework for adapting brand logo colors in dark mode environments. Traditionally, designers have relied on intuitive judgment or experiential heuristics to manually adjust RGB values. In contrast, the perceptual model introduced in this research enables such adjustments to be automated as continuous functional relationships within the CIELAB color space. By standardizing the color adjustment process, the model helps reduce subjective variability among designers and enhances the reproducibility and consistency of brand color representation.

From a brand management perspective, the proposed model also offers a systematic approach to resolving color fragmentation between light and dark modes. Previous studies have demonstrated that color consistency is a key determinant of brand recognition, trust, and consumer preference [20,24]. When brand color identity is compromised, visual credibility may decline, and inconsistencies across platforms can disrupt consumers' memory colors, the colors they subconsciously

associate with familiar brands [25,33]. Therefore, the model proposed in this study can serve as an enterprise-level design management tool, enabling companies to maintain coherent color systems across light and dark modes while improving design workflow efficiency and ensuring brand–UI color consistency.

## Limitations and future work

This study has several limitations that present opportunities for future research and refinement of the color adjustment model.

First, this experiment utilized data from 18 colors. These colors were composed of two tone variations and included 9 distinct hues, with 8 high-chroma colors and 1 gray. The color adjustment model developed in this study was interpolated based on this dataset to generate adjusted colors. As an exploratory, early-stage study, the primary goal was to capture perceptual trends across a diverse hue spectrum rather than to build a finalized or exhaustive model. Therefore, sampling a broad hue range with controlled chroma levels was considered an appropriate and meaningful starting point. However, its effectiveness in adjusting low-chroma colors remains limited. In cases where logos with low chroma are adjusted, the target colors may be derived more from interpolation rather than actual experimental data, potentially leading to greater error in color adjustment. Therefore, future research should consider incorporating a broader range of chroma levels to enhance the precision of the model.

Second, this study considered a black background as a representative dark mode setting. However, in real-world applications, dark mode interfaces often feature varying shades of gray or colored backgrounds [2], which can influence color perception and contrast sensitivity. These background variations may cause inconsistencies in color adjustments [52]. Since the color adjustment vectors in this study were derived under a uniform black background, their applicability to more complex background conditions remains uncertain. This limitation suggests that the current model may require additional refinements to account for perceptual shifts that occur under different dark mode environments.

Third, to reflect the purpose of dark mode [2], participants in the validation phase were encouraged to conduct evaluations under low-light conditions. This setup allowed logos to be assessed while participants' vision was adapted to a dark environment [29,31]. However, dark mode is increasingly being used in various ambient lighting conditions [6]. Whether the adjusted colors validated in this study will perform consistently in brighter environments remains uncertain. This trend in usage context of dark mode suggests that the evaluation of adjusted logo colors should be expanded to consider a broader range of lighting environments for more general applicability.

Fourth, diversifying the participant sample is an important direction for future research. Upcoming experiments should include participants from non-design backgrounds, a wider age range, and diverse cultural contexts to quantitatively examine cross-cultural differences in color perception and brand association. Prior studies have shown that color preferences and brand-related color meanings vary significantly across cultural backgrounds [53,54]. Since the present study included only young Korean participants in their twenties, the findings may not fully account for age-related variations in color perception or cultural influences on color meaning, which limits the generalizability of the results.

Fifth, in the preference survey, participants were instructed to use their own smartphones to test the model's applicability under realistic in-field conditions. Although this approach naturally included a variety of display devices, the color gamut and luminance characteristics of each device were not quantitatively recorded. As a result, it was difficult to precisely determine the color appearance that each participant actually observed. In particular, differences in black level and contrast ratio between OLED and LCD displays may produce nonlinear perceptual effects on color appearance. Future research should therefore investigate these device-dependent variations more systematically through calibrated display measurements.

Another limitation is that the proposed model was evaluated only through a subjective preference survey. While this approach is appropriate for an initial exploratory study, it does not fully capture physiological responses related to visual comfort or strain. Future research should therefore incorporate physiological or neurocognitive measurements to more

objectively validate perceptual comfort. Methods such as electroencephalography to assess visual fatigue, eye-tracking to analyze attention patterns, or pupillometry to measure cognitive load could be combined with subjective evaluations [55]. Such a multi-modal approach would enable a more comprehensive assessment of color adaptation effects in dark mode. Another limitation is that participants' habitual use or preference for light or dark mode was not collected. Since individual familiarity with dark mode can influence both adjustment behavior and subjective evaluation, future studies should include this factor as a covariate or grouping variable to better explain perceptual differences.

By refining the model's accuracy and incorporating a wider range of color stimuli, background conditions, and lighting environments, its practical utility and robustness can be further enhanced.

## Conclusion

This study proposes a color adjustment model that preserves the color characteristics of brand logos while enhancing visual comfort and visibility when transitioning to dark mode. To achieve this, we conducted a color adjustment experiment and analyzed the results to establish principles for color modification. By formulating the transformation as a continuous function in CIELAB, the work situates dark-mode color adaptation within perceptual principles.

Based on these insights, we developed a color adjustment model and applied it to 18 fictitious logos and 18 commercial logos for validation. User evaluations indicated that the adjusted logos were generally preferred over direct dark-mode conversions, particularly for logos that originally used darker colors. While this study serves as an early exploratory step rather than a finalized or universally generalizable solution, it introduces a perceptual approach to dark-mode color adaptation that has been largely overlooked. By grounding color adjustment in human visual cognition instead of arbitrary or purely technical transformations, this work establishes a foundation for future design methodologies and system-level brand color management in dark environments.

## Supporting information

**S1 Code. Python implementation of the color adjustment model.** The Python code and supplementary data for the color adjustment model are openly available at Zenodo (DOI: https://doi.org/10.5281/zenodo.16939055).
(ZIP)

**S1 Table. Raw data of color adjustment experiment.** This file contains original and manually adjusted color values for 18 logo stimuli collected from 31 design-major participants.
(XLSX)

**S2 Table. Raw data of preference survey.** This file includes participant responses on logo color preference and familiarity for both fictitious and commercial logos.
(XLSX)

## Acknowledgments

We thank the students and participants who volunteered in experiment and survey.

## Author contributions

**Conceptualization:** Giyun Lee, Hyeon-Jeong Suk.

**Data curation:** Giyun Lee, Hyeon-Jeong Suk.

**Formal analysis:** Byeongjin Kim, Giyun Lee, Hyeon-Jeong Suk.

**Funding acquisition:** Hyeon-Jeong Suk.

**Investigation:** Giyun Lee.

**Methodology:** Byeongjin Kim, Giyun Lee, Hyeon-Jeong Suk.

**Project administration:** Byeongjin Kim.

**Resources:** Hyeon-Jeong Suk.

**Supervision:** Hyeon-Jeong Suk.

**Validation:** Giyun Lee.

**Visualization:** Giyun Lee.

**Writing – original draft:** Byeongjin Kim, Giyun Lee, Hyeon-Jeong Suk.

**Writing – review & editing:** Byeongjin Kim, Hyeon-Jeong Suk.

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
