## [Decision Letter · Decision Letter 0]

24 Oct 2025

Dear Dr. Suk,

Thank you for submitting your manuscript to PLOS ONE. After careful consideration, we feel that it has merit but does not fully meet PLOS ONE’s publication criteria as it currently stands. Therefore, we invite you to submit a revised version of the manuscript that addresses the points raised during the review process.

We look forward to receiving your revised manuscript.

Kind regards,

Shengqian Sun

Academic Editor

PLOS ONE

Journal Requirements:

2. We note that Figure 2 in your submission contain copyrighted images. All PLOS content is published under the Creative Commons Attribution License (CC BY 4.0), which means that the manuscript, images, and Supporting Information files will be freely available online, and any third party is permitted to access, download, copy, distribute, and use these materials in any way, even commercially, with proper attribution. For more information, see our copyright guidelines: http://journals.plos.org/plosone/s/licenses-and-copyright.

3. We note you have included a table to which you do not refer in the text of your manuscript. Please ensure that you refer to Tables 1 and 3 in your text; if accepted, production will need this reference to link the reader to the Tables.

4. We notice that your supplementary files are included in the manuscript file. Please remove them and upload them with the file type 'Supporting Information'. Please ensure that each Supporting Information file has a legend listed in the manuscript after the references list.

5. Please upload a copy of Supporting Information Figure/Table/etc. which you refer to in your text on page 17.

Reviewers' comments:

Reviewer's Responses to Questions

**Comments to the Author**

1. Is the manuscript technically sound, and do the data support the conclusions?

Reviewer #1: Yes

Reviewer #2: Partly

Reviewer #3: Partly

2. Has the statistical analysis been performed appropriately and rigorously?

Reviewer #1: Yes

Reviewer #2: Yes

Reviewer #3: No

3. Have the authors made all data underlying the findings in their manuscript fully available?

Reviewer #1: Yes

Reviewer #2: Yes

Reviewer #3: No

4. Is the manuscript presented in an intelligible fashion and written in standard English?

Reviewer #1: Yes

Reviewer #2: Yes

Reviewer #3: Yes

Reviewer #1: Thank you for giving me the opportunity to review this manuscript. I think that the topic is interesting. The paper is well structured and easy to read. However, the paper has some flaws that authors must pay attention before the paper can get accepted:

Introduction

This section provides a thorough description of the "popularity of dark modes" and the "current research status of contrast, visibility and readability", but the transition from prior work to the research gap is not sufficiently clear. Please state the research gap and the innovative contribution more explicitly.

At present the manuscript mixes multiple motivations, including brand visual identity, interface comfort, and young user preferences, which makes it difficult for readers to distinguish the study’s theoretical contribution from its applied motivations. I recommend clarifying the hierarchy between the industry context and the theoretical background.

Materials and Methods

The Methods section describes the experimental workflow in reasonable detail; however, it lacks essential details on key control variables and measurement conditions.

The manuscript reports only the participant counts (e.g., 31 in the color adjustment experiment and 89 in the preference test) but lacks demographic and professional background details.

Although the ethics approval number is provided in the manuscript, I recommend moving this information to the beginning of the Methods section or the Participants subsection.

Results

The theoretical rationale and the validation procedure for the model need clearer articulation. In “Building a Color Adjustment Model for Dark Mode,” the authors use Kriging interpolation and CIELAB coordinates; please explain the theoretical motivation for these algorithmic choices and specify the error validation methods to make the modeling section more rigorous and assessable.

The data analysis methods appear somewhat weak. Please add details of the statistical tests and the significance levels to strengthen the credibility of the results.

Discussion

At present the Discussion section, particularly the beginning of “Findings and Insights,” moves directly into restating and comparing results, and it lacks a clear logical progression from experimental findings to theoretical interpretation to practical implications.

Although the manuscript states that “previous studies … focused on optimizing contrast,” it lacks comparative dimensions. For example, did prior work focus only on text readability? Has anyone attempted hue or saturation correction? Please expand the comparative discussion.

While the conclusion notes that the model could be extended to broader UI design, the outlook should be made more concrete. For example, discuss adaptation for animated icons, AR/VR environments, and differences between OLED and LCD displays; and potential contributions to sustainable branding systems or automated color correction systems. This would enhance the paper’s forward-looking value and citation potential.

Limitations and Future Work

The manuscript notes only that the experiments used 18 colors, a black background, and low-light conditions, but it lacks reflection on the diversity of samples and stimuli. For example, the participant pool appears to consist mainly of design students, which may limit generalizability, and cultural differences and device variability (display brands and gamut differences) are not considered.

At present the manuscript only makes a general suggestion to broaden the color range and lighting conditions. Please strengthen the Future Work section with actionable and forward-looking plans.

Conclusion

The conclusion is overly general, as in “user feedback was dominantly positive.” It should focus more tightly on the main findings supported by empirical evidence.

To strengthen the summary of theoretical and practical contributions, please add a sentence in the Conclusion that elevates the study’s value from a methodological contribution to a contribution to design theory.

Reviewer #2: This study gives a systematic, data-supported, and stable method. It helps designers and companies change brand logo colors well when they switch to dark mode. This method helps keep the brand look consistent, makes it comfortable for users to see, and makes sure it is visible.

But there are still things to improve. The study is an early quantitative (number-based) study. As the authors said in the limitations section, it has weaknesses. These include a limited number of color samples, using only one simple background, and limited lighting conditions. Also, a weakness the authors did not mention is that the test subjects were all very similar. Later, the study used an online survey to get user feedback. This is not enough for visual feedback research. Usually, this kind of research uses physiological feedback surveys, like visual neuroscience and EEG. Future follow-up studies can first look for related research methods in other papers and use those as a basis for getting user feedback.

Reviewer #3: ## Reviewer abstract

Authors present a study on developing a computational model to adjust the color in dark mode of brand logos originally designed for light mode user interfaces. First, Authors performed an experiment in a controlled environment to collect data on how a set of participants (Design-major students) manually adjusted artificially generated logos into their dark mode variants. Based on these data, Authors implemented a computational model for automatic color adjustment, which was then tested by a different set of participants on a less controlled experiment. According to their analyses, Authors state that the computational model was well received and that it could be applied across diverse usage contexts.

## Overall comments

I found the topic of the study interesting and timely, given the recent rise in popularity of dark mode user interface and a personal preference for this mode. Overall, the article is well written in terms of grammar and spelling. Concerning the data collection during both experiments, I appreciated the fact that the study had a formal IRB approval, and a few important aspects were taken into consideration in trying to minimize bias; albeit I found not asking participants their preferred mode (light or dark) before the experiments an oversight. The construction of the computational model is also well detailed, and the provided code offers a reference implementation for interested Readers. Nevertheless, the scarce quantification of participant data, the lack of statistical rigor in the (few) conducted analyses, and the unavailability of the most crucial datasets for a perception-based model —i.e., participant-level data— severely hinder the manuscript assertions, and by consequence its publication. In addition, I found that there is a lack of discussion regarding current branding practices for the different modes. For instance, the Netflix logo achieves similar contrast ratios with the same colors in both light and dark modes (over 4.4:1), and its guidelines¹ require a minimum contrast ratio with any background (2.25:1). Finally, there are several presentational issues, including with Figures, which given the very visual-focused nature of the study, I found unfortunate.

## Inadequacy of provided datasets and code

PLOS journals require Authors to make available all data necessary to replicate study findings. This is not the case for the study at hand, as the related code and data uploaded to Zenodo only cover the computational model itself but not the data used to construct such model. There is no individual-level data on how each participant adjusted the logo colors in Experiment 1, which represent the very basis for the proposed perception-based adjustment model. There is no individual-level data on how each participant expressed their logo remembrance and preference in Experiment 2. There is no (anonymized) data regarding individual characteristics of participants. These data are crucial to be able to reproduce Authors’ results and to give support to their assertions and overall discussion in the manuscript. In addition, the code provided (in the form of a Jupyter notebook) should be better prepared as accompanying material to the manuscript in an international context. There are many comments in Korean, and there are also several warnings at stderr level at the end of the notebook.

## Lack of rigor in statistical analysis

Many times, Authors’ statements are not backed by statistical analyses or summaries. For instance, in the section of “Trend of Color Adjustments for Dark Mode Display”, there is no quantification of the changes in L*, C*, and h*, only a textual description is given. In addition, it is not stated if these changes are significant or not. For instance, in line 247 Authors state «The process of adjusting logo colors on a black background resulted in bright colors becoming darker and dark colors becoming brighter, while the hue angle remained relatively stable.» However, this is not quantified, e.g., by giving the average difference in luminance and hue for the two groups (light and dark colors); no statistical test is conducted to verify if this luminance difference between the two groups is significant; and no table nor figure illustrate such differences. This is also important for changes in contrast ratios (mentioned in the Discussion, lines 424 and 429), which again, are not quantified nor tested for significance. Similar comments can be made for the other color dimensions.

In fact, there are only two statistical tests in the study, both of which regard Spearman rank correlation for Experiment 2 and both of which are —in my opinion—not adequate. The first test, on line 407, concerns proportions of logo remembrance and adjustment preference. The correlation is actually moderate (ρ = -.42) and the p-value is relatively close to α = .05. Nonetheless, testing the significance of overall proportions is not adequate for the question at hand (is logo remembrance associated with adjustment preference?). A better approach is to construct a 2×2 contingency table for participants, with “remembrance” (yes/no) and “preference” (non-adjusted/adjusted) and then conduct an association test (χ² or Fisher’s exact test). For the second test, on line 416, which concerns the correlation between color change (measured as the Euclidian distance) and logo preference (non-adjusted/adjusted), perhaps is better to conduct a point-biserial correlation or another test for dichotomous and continuous variables.

## Presentational issues:

- Attention should be paid to PLOS One guidelines² on figures. In many cases the labeling font size is too small. In addition, most of the Figures should be (ideally) in vector format (e.g., EPS) or, if in raster format (e.g., TIFF), they should be a least 300 DPI.

- In Tables 1 and 2, I suggest using white for the name of dark colors as the cell contrast is too low, especially for Dark Blue and Dark Purple. I also suggest a right justification of the numeric cells to ease comparisons, as well as separating the standard deviations in table 2 into their own columns for the same reason.

- In Fig. 4, I suggest annotating the dark mode images (A and B) with the respective HSV values of the Samsung logo, so Readers can have a quantitative notion of how color values change after an adjustment.

- In Fig. 8, plot subtitles erroneously refer to the proportion of participants who preferred the adjusted logos as “ratio”. A ratio is a quantitative relation between two amounts, such as the contrast ratio between two colors.

- Supplemental Figures S1 and S2 are not original, based on a reverse image search. I invite Authors to double check if their supplemental figures respect PLOS One requirements concerning image licensing and copyright.

## Footnotes:

1. https://brand.netflix.com/en/assets/logos/

2. https://journals.plos.org/plosone/s/figures

**Do you want your identity to be public for this peer review?** For information about this choice, including consent withdrawal, please see our Privacy Policy

Reviewer #1: No

Reviewer #2: **Yes: ** Wei-Te, Tsai

Reviewer #3: No

---

## [Author Response · Author response to Decision Letter 1]

19 Nov 2025

We are deeply grateful to the reviewers for their insightful and constructive comments. We are encouraged by the positive reception of the overall framework, methodology, and contributions of our work. The reviewers’ thoughtful suggestions have been invaluable in refining and improving the manuscript. In response, we have made careful revisions to better contextualize the study, enhance methodological transparency, and provide additional details where needed. We believe these revisions have strengthened the clarity and academic rigor of the paper, and we sincerely hope the updated manuscript meets the reviewers’ expectations.

A detailed, point-by-point response is provided in the uploaded document.

---

## [Decision Letter · Decision Letter 1]

7 Dec 2025

Color adjustment of brand logos for dark mode display

PONE-D-25-46843R1

Dear Dr. Suk,

We’re pleased to inform you that your manuscript has been judged scientifically suitable for publication and will be formally accepted for publication once it meets all outstanding technical requirements.

Kind regards,

Shengqian Sun

Academic Editor

PLOS One

Additional Editor Comments (optional):

Reviewers' comments:

Reviewer's Responses to Questions

**Comments to the Author**

Reviewer #1: (No Response)

Reviewer #2: All comments have been addressed

2. Is the manuscript technically sound, and do the data support the conclusions?

Reviewer #1: (No Response)

Reviewer #2: Yes

3. Has the statistical analysis been performed appropriately and rigorously?

Reviewer #1: (No Response)

Reviewer #2: Yes

4. Have the authors made all data underlying the findings in their manuscript fully available?

Reviewer #1: (No Response)

Reviewer #2: Yes

5. Is the manuscript presented in an intelligible fashion and written in standard English?

Reviewer #1: (No Response)

Reviewer #2: Yes

Reviewer #1: (No Response)

Reviewer #2: The revised paper has improved a lot. The authors answered the reviewers' comments very carefully. By adding statistical analysis (like the Chi-square test) and clarifying the participants' backgrounds, you have made the study much stronger and more rigorous.

The proposed CIELAB color model effectively solves the problem of visual comfort and brand identity in dark mode using Kriging interpolation. This work has great value for both academic research and practical design application.

Also, sharing your raw data and Python code is a great step. It makes your research transparent and easier for others to reproduce. Overall, It makes a real contribution to the UI/UX design field.

**Do you want your identity to be public for this peer review?** For information about this choice, including consent withdrawal, please see our Privacy Policy

Reviewer #1: No

Reviewer #2: No

---

## [Editor Report · Acceptance letter]

PONE-D-25-46843R1

PLOS One

Dear Dr. Suk,

I'm pleased to inform you that your manuscript has been deemed suitable for publication in PLOS One. Congratulations! Your manuscript is now being handed over to our production team.

Kind regards,

on behalf of

Dr. Shengqian Sun

Academic Editor

PLOS One